# Lipopolysaccharide from the Cyanobacterium *Geitlerinema* sp. Induces Neutrophil Infiltration and Lung Inflammation

**DOI:** 10.3390/toxins14040267

**Published:** 2022-04-09

**Authors:** Julie A. Swartzendruber, Rosalinda Monroy Del Toro, Ryan Incrocci, Nessa Seangmany, Joshua R. Gurr, Alejandro M. S. Mayer, Philip G. Williams, Michelle Swanson-Mungerson

**Affiliations:** 1Department of Microbiology and Immunology, College of Graduate Studies, Midwestern University, Downers Grove, IL 60515, USA; jswart@midwestern.edu (J.A.S.); rdelto@midwestern.edu (R.M.D.T.); rincro@midwestern.edu (R.I.); 2Biomedical Sciences Program, College of Graduate Studies, Midwestern University, Downers Grove, IL 60515, USA; nseangmany59@midwestern.edu; 3Department of Chemistry, University of Hawaii-Manoa, Honolulu, HI 96822, USA; gurr@amryis.com (J.R.G.); philipwi@hawaii.edu (P.G.W.); 4Department of Pharmacology, College of Graduate Studies, Midwestern University, Downers Grove, IL 60515, USA; amayer@midwestern.edu

**Keywords:** cyanobacteria, lipopolysaccharide, lung inflammation, neutrophils, glucocorticoid-resistant asthma

## Abstract

Glucocorticoid-resistant asthma, which predominates with neutrophils instead of eosinophils, is an increasing health concern. One potential source for the induction of neutrophil-predominant asthma is aerosolized lipopolysaccharide (LPS). Cyanobacteria have recently caused significant tidal blooms, and aerosolized cyanobacterial LPS has been detected near the cyanobacterial overgrowth. We hypothesized that cyanobacterial LPS contributes to lung inflammation by increasing factors that promote lung inflammation and neutrophil recruitment. To test this hypothesis, c57Bl/6 mice were exposed intranasally to LPS from the cyanobacterium member, *Geitlerinema* sp., in vivo to assess neutrophil infiltration and the production of pro-inflammatory cytokines and chemokines from the bronchoalveolar fluid by ELISA. Additionally, we exposed the airway epithelial cell line, A549, to *Geitlerinema* sp. LPS in vitro to confirm that airway epithelial cells were stimulated by this LPS to increase cytokine production and the expression of the adhesion molecule, ICAM-1. Our data demonstrate that *Geitlerinema* sp. LPS induces lung neutrophil infiltration, the production of pro-inflammatory cytokines such as Interleukin (IL)-6, Tumor necrosis factor-alpha, and Interferongamma as well as the chemokines IL-8 and RANTES. Additionally, we demonstrate that *Geitlerinema* sp. LPS directly activates airway epithelial cells to produce pro-inflammatory cytokines and the adhesion molecule, Intercellular Adhesion Molecule-1 (ICAM-1), in vitro using the airway epithelial cell line, A549. Based on our findings that use *Geitlerinema* sp. LPS as a model system, the data indicate that cyanobacteria LPS may contribute to the development of glucocorticoid-resistant asthma seen near water sources that contain high levels of cyanobacteria.

## 1. Introduction

Asthma is a chronic inflammatory condition that undermines the airways. A common misconception is that asthma is exclusively caused by T helper cell 2-driven eosinophil responses. While Th2-driven asthma predominates, up to 10% of individuals with asthma symptoms are glucocorticoid-resistant and their symptoms are independent of eosinophilic activation [1,2,3]. While the percentage of these patients is relatively low, they account disproportionately for 50–80% of health care costs since they have limited treatment options [4,5,6]. While glucocorticoid-resistant asthma is a heterogenous disorder, a source of asthma-like symptoms can be induced by another immune cell of the immune system, the neutrophil [1,7]. Non-eosinophilic asthma is strongly associated with neutrophil recruitment and activation and innate immune responses activated by Toll-like receptors (TLR), such as TLR2, TLR4, and NLRP3 [3], and activation of Th17 cells [8]. TLR4 is strongly activated by bacterial lipopolysaccharide (LPS) and previous studies demonstrate that bacterial lipopolysaccharide (LPS) specifically induces neutrophil-based lung inflammation and asthma [9,10]. Furthermore, a study of individuals with glucocorticoid-resistant asthma demonstrate that these individuals have a signature of classic LPS activation and high levels of endotoxins within bronchial alveolar lavage (BAL) fluid [11]. The inhalation and accumulation of LPS can lead to neutrophil influx into the lungs where these cells can release destructive enzymes and pro-inflammatory cytokines that result in lung immunopathology [10,12].

LPS can be derived from many bacterial sources, but a bacterium of increasing concern is cyanobacteria [9,13,14]. This group of bacteria are a major cause of toxic freshwater blooms that can be ingested through contaminated water sources and inhaled through aerosols [9,15]. Due to the potential impact on public health, cyanobacterial blooms result in public health warnings, the closing of beaches, and necessary decontamination of drinking water by the Environmental Protection Agency (EPA) and the Centers for Disease Control (CDC) [16,17]. Furthermore, more recent evidence suggests that cyanobacteria are detected at distances away from outdoor water blooms [9], and even the indoor environment of an office building [18]. Importantly, these blooms result in respiratory, gastrointestinal, and systemic symptoms, such as fever [14,19]. The ability of cyanobacteria to induce respiratory disease was unclear until recent advancements demonstrated that cyanobacteria and their toxins not only can become aerosolized [9,15,16,20], but that cyanobacteria can be routinely found in the upper respiratory tract and central airways [20]. Even more significantly, cyanobacterial endotoxins, such as LPS, have been found aerosolized near coastal areas [21].

The respiratory disease caused by cyanobacteria described above could be induced through cyanobacteria toxins, including LPS and/or cyanotoxins. Cyanotoxins have previously been demonstrated to induce pulmonary thrombosis [22] after injection and crude extracts from *Microcystis aeruginosa* induced neutrophil infiltration into the lungs after intraperitoneal injection [23]. However, based on the finding that LPS is aerosolized near coastal areas [21], we hypothesized that LPS-mediated activation of the immune system in the respiratory tract [24] could also promote cyanobacterial-induced respiratory disease. In support of this statement, one report indicates that endotoxins from cyanobacteria can contribute to an acute respiratory response after aerosol inhalation [19]. However, the mechanism by which these endotoxins mediate this effect is unknown. Previous in vitro work from our laboratory indicates that the endotoxin LPS from a cyanobacterium belonging to the genus *Geitlerinema* (previously included in the genus *Oscillatoria*, but was reclassified as *Geitlerinema*, within the same order Oscilatoriales, after 16 s phylogenetic analysis) activates multiple immune cells. Even though *Geitlerinema* sp. is not a prototypical bloom-forming cyanobacterial species, it is known to be present in blooms [25,26,27], and it has served as a model system for obtaining sufficient quantities of cyanobacteria LPS for our studies. *Geitlerinema* sp. LPS activates both murine B cells [28] and induces pro-inflammatory cytokine production by human monocytes [29]. Our data further demonstrated that *Geitlerinema* sp. LPS does this by stimulating Toll-like receptor 4 (TLR4) in a manner that is similar, yet distinct from the positive control, *Escherichia coli* (*E. coli*) LPS [28]. Our data indicate that *Geitlerinema* sp. LPS induces monocytes to produce IL-1, IL-6, and TNF-alpha [29], which are not only important in promoting systemic symptoms, such as fever, but also are important in inducing lung inflammation [1].

Based on these previous data, we hypothesized that cyanobacterial LPS are responsible for promoting the asthma-like symptoms seen in individuals near cyanobacterial blooms. To test this hypothesis, we exposed mice to *Geitlerinema* sp. LPS intranasally in a well-established model of LPS-induced lung inflammation that analyzes inflammation (as determined by leukocyte infiltration), cytokine/chemokine production, and induction of adhesion molecules by airway epithelial cells.

## 2. Results

### 2.1. Aerosolized Geitlerinema sp. LPS Induces Lung Inflammation In Vivo

To determine if cyanobacteria lipopolysaccharide induces neutrophil-dependent lung inflammation, mice were exposed to *Geitlerinema* sp. LPS intranasally and the lungs were examined for inflammation 24 h later by hematoxylin and eosin staining, as described previously for *E. coli* LPS [12,30]. For all experiments, *E. coli* LPS was used as a positive control for inducing lung inflammation and leukocyte infiltration [12]. As shown in Figure 1A, the lungs of *E. coli* LPS showed significant leukocyte infiltration when compared to mice that were only exposed to PBS. When the lungs of mice exposed to *Geitlerinema* sp. LPS were analyzed for inflammation, the lungs of these mice also demonstrate an influx of leukocytes in comparison to mice exposed to PBS only (Figure 1A). However, the *Geitlerinema* sp. LPS did not induce a level of leukocyte influx that was equivalent to *E. coli* sp. LPS (Figure 1A).

Since *Geitlerinema* sp. LPS-dependent leukocyte inflammation was detected by hematoxylin and eosin, we sought to determine the composition of infiltrating cells. Therefore, we isolated the bronchoalveolar lavage fluid (BAL) 24 h after LPS exposure and stained the cells from the BAL using a Diffquick stain that allows for the differentiation among cell types. As shown in Figure 1B, the predominant cell in the BAL are macrophages, with very few neutrophils, eosinophils and lymphocytes in non-LPS-exposed mice. In contrast, in both *E. coli* and *Geitlerinema* sp. LPS-exposed mice, there is a significant increase in the percentage of inflammatory neutrophils and a decrease in the percentage of macrophages in the lungs of these mice (Figure 1B). While we observed no differences in the types of cells infiltrating the lungs when exposed to either *E. coli* or *Geitlerinema* sp. LPS (Figure 1B), we still saw a decrease in overall inflammation and cellular infiltrate in the lungs by H and E staining when comparing mice exposed to *Geitlerinema* sp. LPS versus *E. coli* LPS (Figure 1A). Therefore, we quantified the actual number of infiltrating neutrophils in the lungs of PBS and LPS-treated mice using the DiffQuick stained slides. As shown in Figure 1C, *E. coli* LPS induced a dramatic increase in the number of neutrophils in the BAL in comparison to mice exposed to PBS alone. When the number of neutrophils was calculated in the BAL of *Geitlerinema* sp. LPS-treated mice, the numbers were significantly higher than PBS alone, but not to the level of *E. coli* LPS (Figure 1C), which are consistent with the level of infiltration seen by H and E staining in Figure 1A. Finally, we used flow cytometry to confirm the influx of neutrophils and the decrease in macrophages by analyzing the cells present in the BAL. The number of neutrophils (CD11b + Ly-6G+ cells) were significantly increased in both *E. coli* and *Geitlerinema* sp. LPS-treated mice with a concomitant decrease in F4/80 + CD11b+ macrophages (Figure 1D). Taken together, these data demonstrate that *Geitlerinema* sp. LPS induces neutrophil infiltration and lung inflammation in vivo.

### 2.2. Geitlerinema sp. LPS Induces Cytokine Production in the Lungs of LPS-Treated Mice In Vivo

The mechanism by which neutrophils are recruited to the lungs after LPS exposure requires the production of chemokines produced by airway epithelial cells [32,33]. Based on our data in Figure 1B–D that demonstrated an increase in neutrophil recruitment, we expected that there would be an increase in the neutrophil-recruiting chemokine IL-8 (the murine equivalent of IL-8 is called the keratinocyte-derived chemokine ( KC) [34]) in LPS-treated mice. However, we also tested the presence of other chemokines in the BAL of *Geitlerinema* sp. LPS-treated mice. As shown in Figure 2A, *Geitlerinema* sp. LPS significantly increased the levels of KC (the murine homolog of IL-8) and RANTES, with no significant increase in MIP-2 in the BAL.

Within the lung of *E. coli* LPS-treated mice, a very strong pro-inflammatory cytokine response is induced [12]. To determine if *Geitlerinema* sp. LPS induces a pro-inflammatory cytokine response versus an anti-inflammatory immune response, we analyzed cytokine levels within the BAL. As shown in Figure 2B, *Geitlerinema* sp. LPS induced IL-6, TNF-alpha and IFN-gamma, albeit at lower levels than *E. coli* LPS. In contrast, *Geitlerinema* sp. LPS did not induce IL-17 production, versus *E. coli* LPS which was capable of inducing IL-17 (Figure 2B). Finally, while *E. coli* LPS slightly induced the production of the anti-inflammatory cytokine IL-10, *Geitlerinema* sp. LPS does not increase IL-10 production (Figure 2C).

### 2.3. Geitlerinema sp. LPS Induces ICAM-1 Levels in the Lungs of LPS-Treated Mice In Vivo and Human Airway Epithelial Cells In Vitro

Once neutrophils are recruited to the lungs by chemokines such as RANTES and IL-8, neutrophils must adhere to airway endothelial cells through an interaction of LFA (Leukocyte Function-associated Antigen) on the neutrophil with ICAM-1 on airway epithelial cells to leave the circulation and enter the lungs. To confirm that *Geitlerinema* sp. LPS increased ICAM-1 on airway epithelial cells in vivo, the lungs from either *Geitlerinema* sp. LPS or *E. coli* LPS-exposed mice were harvested after 24 h and stained for ICAM-1 levels through immunohistochemistry. As shown in Figure 3A, airway epithelial cells from lungs of mice exposed to either *Geitlerinema* sp. or *E. coli* LPS show greater ICAM-1 levels as determined by increased brown staining of the cells. These data indicate that *Geitlerinema* sp. LPS not only increases the production of chemokines to recruit neutrophils but also upregulates adhesion molecules that promote the extravasation of neutrophils from the blood to the airways in the lung.

Since the previous in vivo experiments were performed in mice, we wanted to confirm that that *Geitlerinema* sp. LPS increases ICAM-1 levels on human airway epithelial cells. Therefore, the human airway epithelial cell line, A549, was exposed to either *Geitlerinema* sp. LPS or *E. coli* LPS for 6 h in vitro and analyzed for ICAM-1 levels by immunofluorescence. Exposure of these cell lines to *Geitlerinema* sp. LPS increased ICAM-1 levels on the A549 human airway epithelial cell line in comparison to untreated cells (Figure 3B). Taken together, these data indicate that *Geitlerinema* sp. LPS directly activates airway epithelial cells in vitro to induce an environment that promotes the extravasation of neutrophils into the lungs and cause pathology in vivo.

### 2.4. Geitlerinema sp. LPS Induces IL-6 and IL-8 in an NF-kB-Dependent Manner In Vitro

Finally, we sought to confirm that *Geitlerinema* sp. LPS directly induces human airway epithelial cell lines to upregulate pro-inflammatory cytokines, such as IL-6, and chemokines, such as IL-8, as seen in our murine in vivo model. A549 cells were exposed to increasing concentrations of *Geitlerinema* sp. LPS with *E. coli* LPS as a positive control in vitro. After 24 h, we analyzed IL-6 and IL-8 levels by ELISA. As shown in Figure 4A,B, *Geitlerinema* sp. LPS showed a significant increase in both IL-6 and IL-8 in the A549 cell line at the highest concentration of *Geitlerinema* sp. LPS. Additionally, our laboratories have previously demonstrated that *Geitlerinema* sp. LPS induces NF-kB activation [28]. Since both IL-6 and IL-8 are induced by NF-kB activation [35,36], we tested whether the *Geitlerinema* sp. LPS-dependent increase in IL-6 and IL-8 is NF-kB dependent. As shown in Figure 4C,D, the addition of the NF-kB inhibitor, Bay 11-7085 blocks the *Geitlerinema* sp. LPS-dependent increase in both IL-6 and IL-8, demonstrating that *Geitlerinema* sp. LPS acts in an NF-kB-dependent manner.

## 3. Discussion

These data are the first to demonstrate that a cyanobacteria LPS, namely *Geitlerinema* sp. LPS induces neutrophil influx and lung inflammation in vivo. While *Geitlerinema* sp. is not a typical participant in cyanobacterial blooms, it serves as a useful experimental system since we are able to produce and purify high levels of LPS (>90%). Due to the recent increase in cyanobacterial blooms [15] and their demonstrated aerosolization [9,15,16], these data have significant implications for the impact of cyanobacteria on human respiratory disease. Importantly, previous prospective cohort studies have demonstrated that study participants that were exposed to cyanobacterial blooms were 2.1 (95% CI: 1.1–4) times as likely to demonstrate respiratory symptoms [37].

Our data are the first to demonstrate that cyanobacteria LPS may contribute to neutrophil-dependent lung inflammation. Cyanobacteria blooms are increasingly on the rise due to global warming [38,39] and therefore, our exposure to these bacteria and their biological products will concomitantly rise. A recent study indicates that cyanobacteria are commonly found in both the upper and lower respiratory tracts in humans [20]. In this study the authors demonstrated using cyanobacteria-specific PCR amplification that cyanobacteria 16s rDNA was found in 92% of the upper respiratory tracts and 79% of the bronchoalveolar fluid of the individuals tested [20]. The findings from this previous study indicate that the humans are potentially chronically exposed to cyanobacteria in the airways, which makes the data from this study even more pertinent.

Cyanobacteria produce multiple toxins, such as microcystins, saxitoxins, and anatoxins [15]. *Oscillatoria* is the most common genus of cyanotoxin producers, but historically those strains were identified using morphological features, and subsequent phylogenetic analysis has resulted in the reclassification of many *Oscillatoria* strains, like the *Geitlerinema* sp. used in this study, into new genera within the order Oscillatoriales further deepening our understanding of the diversity within cyanobacterial blooms. To date, *Geitlerinema* strains have been reported in several freshwater environments [40,41,42] and been shown to produce microcystins [41] and other cyanotoxins [40,43]. Previous data have analyzed the effects of cyanobacteria toxins, especially microcystins, for overall in vivo toxicity [15]. For example, microcystins induce the apoptosis of lymphocytes [44,45], increased the production of pro-inflammatory cytokines by macrophages [46,47,48], and also enhanced the ability of LPS-exposed macrophages to produce nitrous oxide and pro-inflammatory cytokines [46,49]. The finding that microcystins could augment LPS-mediated immune cell activation is interesting in light of the fact that many cyanobacteria produce both LPS and microcystins. Studies that analyze the impact of these two important toxins on airway inflammation are important to pursue as a future direction.

The idea that multiple toxins and/or cyanobacterial strains can have synergistic effects is not novel. One previous study demonstrated that crude extracts from cyanobacteria can act as allergens in a dermatitis model [50]. Interestingly, these data demonstrate allergenic differences among cyanobacterial strains and that combinations of cyanobacterial extracts demonstrate a synergistic impact when both cyanobacterial strain allergens are applied at the same time [50]. Our data indicate that a single cyanobacterial species LPS is capable of inducing lung inflammation, but since individuals are exposed to and harbor multiple cyanobacteria species [15], it will be important to test the contributions of multiple cyanobacteria species concomitantly in our mouse model.

In line with previous human and murine studies [12,30,32], our data indicate that *Geitlerinema* sp. LPS induces pro-inflammatory cytokines after intratracheal exposure. Of note, one cytokine that has been implicated in both animal and human studies, IL-17, was not induced by *Geitlerinema* sp. LPS. However, differences in experimental timing and concentration of LPS used in these studies could influence the differences in our system. We have repeatedly shown that *Geitlerinema* sp. LPS requires higher concentrations of LPS to induce cytokine production and immune cell activation when compared to *E. coli* LPS [28,29,51]. Therefore, we would propose that if higher levels of *Geitlerinema* sp. LPS were used in our experiments that we would induce higher levels of all pro-inflammatory cytokines, including IL-17. Alternatively, concomitant challenge of *Geitlerinema* sp. LPS with allergen may be required to induce IL-17 production, similar to previous studies with *E. coli* LPS [52].

The current study is consistent with previous findings in regard to the fact that *Geitlerinema* sp. LPS is not as potent as *E. coli* LPS with respect to its ability to induce immune cell activation [28,29] This may be due to the fact that *Geitlerinema* sp. LPS is unable to activate the IRF3 pathway after TLR4 engagement, in contrast to *E. coli* LPS that activates both the NF-kB and IRF3 pathways [28].Based on these previous in vitro findings, it was possible that *Geitlerinema* sp. LPS would not be sufficiently potent to induce lung inflammation in vivo. However, the results presented here are the first to demonstrate the ability of *Geitlerinema* sp. LPS to induce inflammation and immune cell activation in vivo. Furthermore, our data are indicative of an acute model. It is likely that cyanobacteria aerosol exposure is chronic [20], and therefore future studies that analyze the impact of chronic cyanobacterial LPS exposure is worth pursuing.

Epidemiology studies over the past two decades have significantly contributed to our understanding of cyanobacterial exposure on the health of both humans and animals [9,15,16,17,21,53,54,55,56]. We now have evidence that we are constantly exposed to cyanobacteria and that these organisms, as well as their cellular products, negatively impact our health. Our current findings provide a mechanism by which cyanobacteria and their LPS contribute to airway inflammation and human disease, as has been implicated recently (18). Future studies that analyze the LPS from additional cyanobacteria in coordination with additional epidemiology studies will advance our understanding of cyanobacteria-mediated lung inflammation, and potentially identify cyanobacteria as a contributing factor to glucocorticoid-resistant asthma. These findings can be used to either identify treatments that target cyanobacterial-generated disease or emphasize the necessity of public health policies that decrease cyanobacterial exposure.

In conclusion, our data demonstrate that *Geitlerinema* sp. LPS induces neutrophil-dependent lung inflammation and induces both pro-inflammatory cytokines, chemokines, and upregulates adhesion molecules on airway epithelial cells. We further determined that *Geitlerinema* sp. LPS induces human airway epithelial cells in vitro to upregulate the pro-inflammatory cytokine IL-6 and the chemokine IL-8 in an NF-kB-dependent manner. Taken together, these data demonstrate that aerosolized cyanobacteria LPS could contribute to the incidence of neutrophil-dependent, glucocorticoid-resistant asthma seen among individuals with exposure to cyanobacteria LPS.

## 4. Materials and Methods

### 4.1. Cell Culture

A549 cell line (human lung adenocarcinoma, ATCC CCL-185) was maintained in F-12K media containing 10% fetal bovine serum and 1% Pen-Strep at 37 °C, 5% CO_2_.

### 4.2. Lipopolysaccharide Isolation from HCC1097

LPS from the Hawaii Culture Collection (HCC) (University of Hawaii, Manoa, HI, USA) 1097 strain of *Geitlerinema* sp. (previously referred to in our publications as *Oscillatoria* sp. LPS) was purified as described previously [28,29]. Briefly, the harvested freeze-dried biomass was extracted in accordance with the commonly utilized hot phenol-water extraction (Westphal and Jann, 1965; Papageorgiou et al., 2004). Briefly, up to 1 g of biomass was transferred to a 50 mL centrifuge tube and incubated in 40 mL of 1:1, distilled H_2_O: 90% aq. phenol in a 68 °C water bath for 30 min. After this time, the mixture was vigorously vortexed and subsequently centrifuged at 2500 r.c.f. for 15 min. The aqueous supernatant of the biphasic mixture was collected carefully as to not to remove cellular debris. This procedure was repeated twice more by adding fresh volumes of distilled H_2_O to the extracting centrifuge tube. The pooled extracts were dialyzed overnight in one liter of Milli-Q H_2_O (Millipore-Sigma, Burlington, MA, USA) in a 3000 MWCO (molecular weight-cutoff) dialysis membrane in order to remove residual phenol. This dialysis was repeated a total of three times. The dialyzed extract was freeze-dried. At this stage, the crude LPS was resuspended to a concentration of 5 mg/mL in 0.1 M TRIS-HCl buffer and incubated at room temperature overnight while on an orbital shaker in the presence of 25 μg/mL of RNase. The material was then dialyzed as described previously and freeze-dried and stored at −20 °C for later chemical analysis. The purity of the *Geitlerinema* sp. LPS reached 90% as determined by UV as previously described (25) and was negative for additional cyanobacterial toxin sample by LC-MS. To assess the nucleic acid content of the lipopolysaccharide sample, UV measurements were taken on a BioSpec-nano spectrophotometer (Thermo-Scientific, Waltham, MA, USA). The measured readings at 230, 260, and 280 nm of a 100 μg/mL aqueous solution of purified LPS were used to calculate the mass percentage of dsDNA or RNA. The instrument’s operating software used a different calculation for each nucleic acid type. The final results indicated that the working sample of LPS was approximately 10% nucleic acid. Coomassie-stained SDS-PAGE did not show any significant amounts of protein contamination.

### 4.3. In Vivo LPS Exposure

All animal experiments were reviewed and approved by the Institutional Animal Care and Use Committee at Midwestern University (Approval #3143). C57Bl/6 mice were housed in the American Association of Laboratory Animal Care (AALAC)-accredited Animal Research Facility at Midwestern University (Downers Grove, IL, USA). All mice were housed in temperature-controlled (21.2 °C) rooms under a 12 h light/dark cycle and provided with autoclaved food and deionized water ad libitum. C57Bl/6 mice (4–8-week-old male and female mice approximately 20 g in weight) were treated intranasally with 1 µg/g of *E. coli* LPS (Millipore Sigma, Burlington, MA, USA) suspended in phosphate buffered saline (PBS), 1µg/g of *Geitlerinema* sp. LPS suspended in PBS, or 50 µL of PBS alone (3–5 mice per group). The concentration of LPS used is based on previous in vivo studies [12] and is an among of LPS that induced a 20% or greater decline in lung function in 50% of humans tested [57]. After 24 h, mice were euthanized before bronchial alveolar lavage (BAL) was collected by flushing lungs with 0.8 mL BAL fluid (10% FCS, 1 mM Ethylenediaminetetraacetic acid (EDTA), 1X PBS). Lungs were excised and fixed in formalin for 24 h. BAL fluid (BALF) was counted and cytospun onto slides and assessed for neutrophils and macrophages using Diff-Quick staining (Dade Behring, Deerfield, IL, USA) as previously described [31]. Each in vivo experiment was performed at least twice with 3–5 mice per group in each experiment.

### 4.4. Flow Cytometry

Cells from BAL were spun down and supernatants were collected. Cells were washed in FACS buffer (PBS/1%FCS) and blocked with TruStain FcX™ (anti-mouse CD16/32) antibody (Biolegend, San Diego, CA, USA) for 30 min and washed again in flow cytometry (FACS) staining buffer. They were stained with PE anti-mouse CD11b, APC/Fire™750 anti-mouse F4/80 and Brilliant Violet 510™ anti-mouse Ly-6G (Biolegend, San Diego, CA, USA) for 30 min in the dark on ice. The cells were subsequently washed and analyzed using CytoFLEX flow cytometer (Beckman Coulter, Indianapolis, IN, USA. Flow cytometry compensation was performed using UltraComp eBeads™ compensation beads (Invitrogen, Waltham, MA, USA) per the manufacturer’s instructions. Viable single cells were gated based on forward light scatter and side light scatter. Fluorescence minus one (FMO) controls were used to determine gates for CD11b, F4/80, and Ly-6G positive cells.

### 4.5. MULTIPLEX Assay for Cytokines and Chemokines

Cell supernatants from BALF were plated for analysis using the MILLIPLEX Multiplex Assay (Millipore Sigma, Burlington, MA, USA) according to the manufacturer’s protocol. The following analytes were measured: IFN-gamma, IL-6, IL-10, IL-17a, KC (IL-8), MIP-2, RANTES, and TNF-a. Results were gathered using the MAGPIX plate reader (Millipore Sigma, Burlington, MA, USA) and data analysis was performed using the MILLIPLEX Analyst Software (Millipore Sigma, Burlington, MA, USA).

### 4.6. Hematoxylin and Eosin Staining

Formalin-fixed lung tissues were embedded in paraffin wax and cut into 5 µm sections. Slides were deparaffinized in xylene for 15 min followed by rehydration by transferring through graded alcohols before incubating in 3% hydrogen peroxide for 10 min before incubating in hematoxylin for 2 min. Slides were washed and dipped in ammonia water (0.25%) until sections turned blue. Then, slides were rinsed and dipped in eosin 20× followed by dehydration through graded alcohols, incubated in xylene for 1 min and sealed.

### 4.7. Immunohistochemistry

Formalin-fixed lung tissues were embedded in paraffin wax and cut into 5µm sections. Tissues were mounted on slides and analyzed by immunohistochemistry. Tissues were immunoperoxidase-stained using Vector Elite ABC kits (Vector Laboratories, Burlingame, CA, USA) with DAB (Vector Laboratories, Burlingame, CA, USA) as a chromogen. Slides were deparaffinized in xylene for 15 min followed by rehydration by transferring through graded alcohols before incubating in 3% hydrogen peroxide for 10 min to retrieve antigen. Nonspecific binding to the tissues was blocked by 5% goat serum for 1 h. Slides were incubated at 4 °C overnight in antibodies to rabbit CD54/ICAM-1 (1:25 Cell Signaling #4915, Danvers, MA, USA).

### 4.8. Immunofluorescence Staining

A549 cells (1.5 × 10⁵) were incubated in individual wells of a 24-well plate with coverslips placed in each well for 24 h. *E. coli* and *Geitlerinema* sp. LPS was initially diluted in PBS. Cells were treated with 100,000 ng/mL of *E. coli* LPS or 100,000 ng/mL of *Geitlerinema* LPS for 6 h. This concentration was used based on previous studies demonstrating a peak LPS response of immune cells in vitro [28,29]. Cells were fixed with methanol at 4 °C overnight. Cells were washed three times with PBS-Tween (0.02%) and incubated in 5% normal goat serum for 1 h at room temperature to block nonspecific binding. After subsequent washing, cells were incubated at 4 °C overnight in primary antibodies to rabbit CD54/ICAM-1 (1:25 Cell Signaling, Danvers, MA, USA). Cells were washed of primary antibody, then incubated with anti-rabbit IgG (AlexaFluor 488, Cell Signaling #4412, Danvers, MA, USA) at room temperature for 1 h, washed again, and mounted in VectaShield HardSet™ with DAPI (Vector Laboratories, Burlingame, CA, USA). Images were obtained on a Nikon EclipseTi confocal microscope (Nikon Instruments, Melville, NY, USA). Image files were quantified by the MFI and mean gray value using imageJ software (National Institutes of Health (NIH), Bethesda, MD, USA). The experiment was performed independently three times.

### 4.9. ELISA

A549 cells (7.5 × 10⁴) were seeded in a 96-well plate and incubated for 24 h. All LPS was initially diluted in PBS. Cells were treated with various concentrations of *E. coli* LPS or *Geitlerinema* sp. LPS (0, 1 K, 10 K, 100 K ng/mL) for 24 h. These concentrations used were based on previous studies demonstrating a peak LPS response of immune cells in vitro [28,29]. Supernatants were collected and plated for IL-6 or IL-8 production using a LEGEND MAX™ human IL-6 or IL-8 ELISA kit (Biolegend, San Diego, CA, USA), respectively, according to the manufacturer’s instructions. Bay 11-1075 was purchased from MedChem Express (Monmouth Junction, NJ, USA) and was diluted to a final concentration of 5 uM. Results were obtained using a Multiskan™ FC Microplate Photometer (ThermoFisher, Waltham, MA, USA). The experiment was performed independently three times.

### 4.10. Statistics

Each experiment was performed at least 3 times. For the data in Figure 1 and Figure 2, experiments were first analyzed by a one-way ANOVA followed by a Bonferroni post hoc test. For experiment 4A-B, a 2-way ANOVA was performed, and the data in Figure 4C,D was analyzed using a Student’s *t*-test. All analysis was done using GraphPad Prism Software (La Jolla, CA, USA).

## Figures and Tables

**Figure 1 toxins-14-00267-f001:**
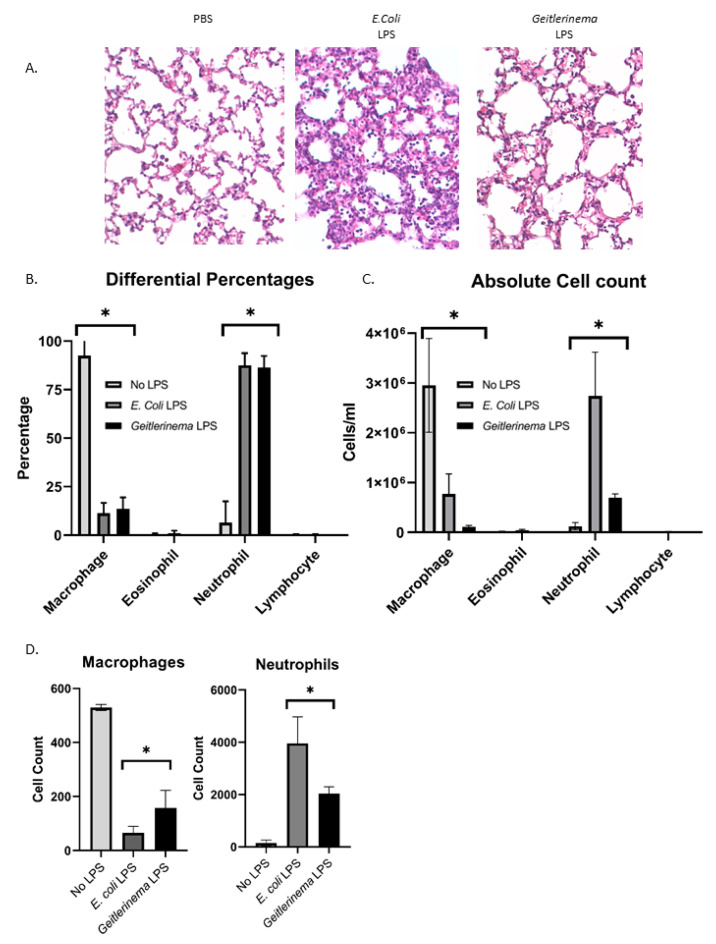
*Geitlerinema* sp. LPS induces neutrophil infiltration and inflammation in the lung. C57Bl/6 mice were treated with either PBS, *E. coli* LPS (1 µg/g) or *Geitlerinema* sp. LPS (1 µg/g) for 24 h. (**A**) Hematoxylin and eosin staining of formalin-fixed lung tissue (**B**) Cells from the BAL were isolated and stained with DiffQuick as described in the Materials and Methods. Differential cell percentages were determined as described previously [31] (**C**) Cell number quantification of cells from the BAL stained with Diffquick (**D**) Flow cytometry to quantify cell number of macrophages (F4/80 + CD11b+) versus neutrophils (Ly-6g+) in the BAL from untreated or LPS-treated mice. In (**A**), the data are representative of and in (**B**) the data are a combination of 3–4 mice. * indicates *p* < 0.05 when compared with PBS treatment.

**Figure 2 toxins-14-00267-f002:**
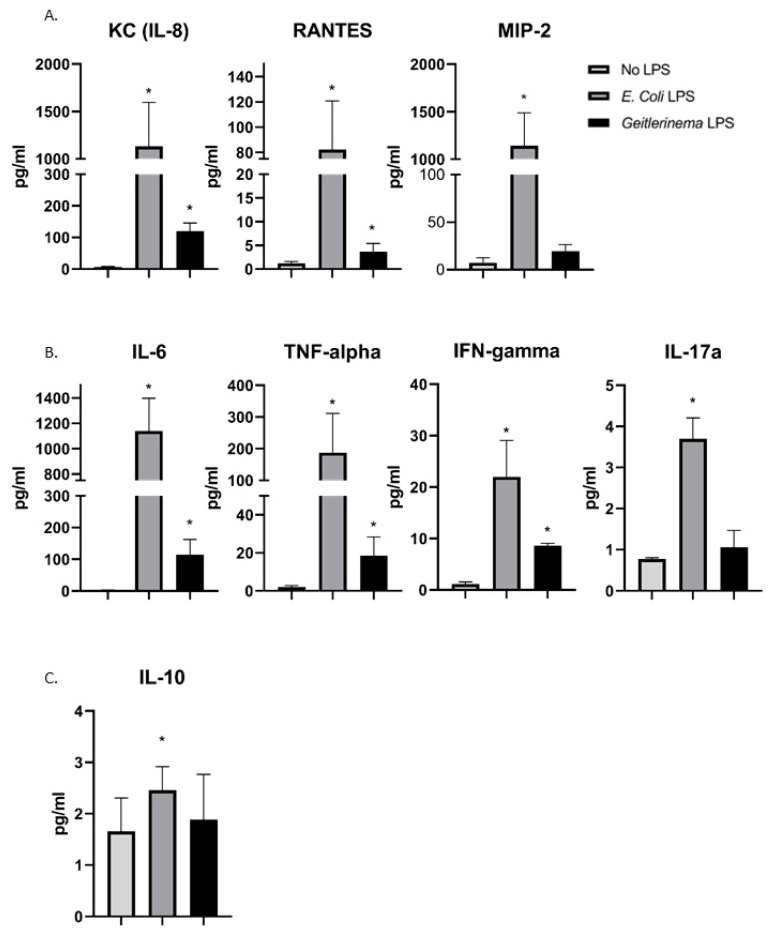
*Geitlerinema* sp. LPS induces cytokine production in the lungs of LPS-treated mice in vivo. C57Bl/6 mice were treated with either PBS, *E. coli* LPS (1 µg/g) or *Geitlerinema* sp. LPS (1 µg/g) for 24 h and BAL was analyzed using a MILLIPLEX Multiplex Assay to analyze (**A**) chemokine production, (**B**) pro-inflammatory cytokine production, or (**C**) anti-inflammatory cytokine production. Data are combined from the BAL of 4–6 mice. * indicates a *p* < 0.05 when compared to PBS treatment. Abbreviations in (**A**–**C**) include: KC = keratinocyte-derived chemokine, RANTES = Regulated on Activation, Normal T cell Expressed and Secreted, MIP-2 = macrophage inflammatory protein-2, IL-6 = interleukin 6, TNF-alpha = tumor necrosis factor alpha, IFN-gamma = interferon gamma, IL-17a = interleukin-17a, and IL-10 = interleukin-10.

**Figure 3 toxins-14-00267-f003:**
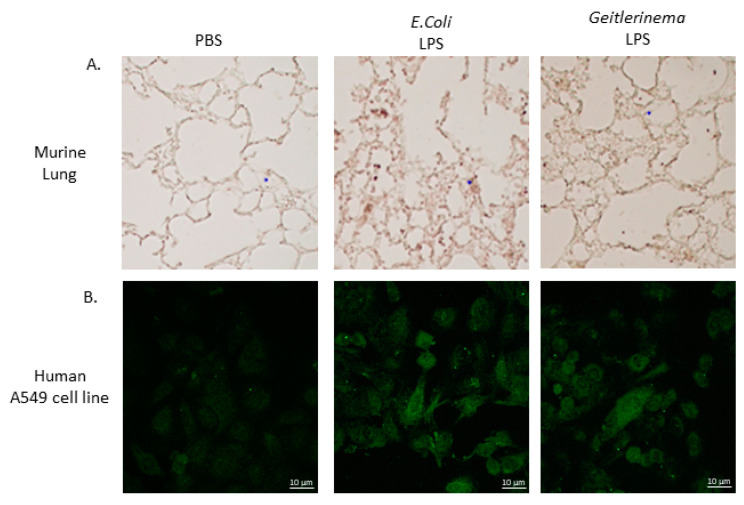
*Geitlerinema* sp. LPS induces ICAM-1 levels in the lungs of LPS-treated mice in vivo and human airway epithelial cells in vitro. (**A**). C57Bl/6 mice were treated with either PBS, *E. coli* LPS (1 µg/g) or *Geitlerinema* sp. LPS (1 µg/g) for 24 h. Lungs were formalin-fixed and embedded in paraffin. Five-micron sections were analyzed by immunohistochemistry at 10× magnification with antibodies specific for ICAM-1 as described in the Materials and Methods. Data are representative of 3 mice (**B**). A549 cells were incubated in the absence of either *Geitlerinema* sp. LPS (100,000 ng/mL) or *E. coli* LPS (100,000 ng/mL) for 6 h and analyzed for ICAM-1 staining by immunofluorescence with magnification at 60×. Data are representative of three experiments with similar results.

**Figure 4 toxins-14-00267-f004:**
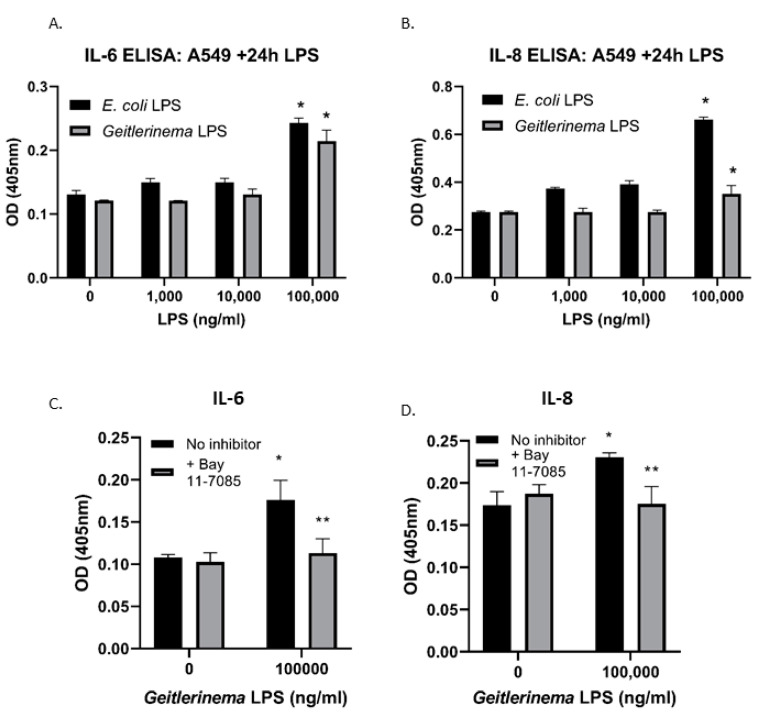
*Geitlerinema* sp. LPS induces IL-6 and IL-8 in an NF-kB-dependent manner in vitro. (**A**,**B**) Airway epithelial cell lines A549 cells were incubated in the presence of increasing concentrations of either *E. coli* LPS (as a positive control) or *Geitlerinema* sp. LPS for 24 h and IL-6 (**A**) or IL-8 (**B**) levels were determined by ELISA. (**C**,**D**) A549 cells were incubated for 24 h in the presence of LPS and in the absence or presence of 5 uM Bay 11-7085 before IL-6 (**C**) or IL-8 (**D**) levels were determined by ELISA. Data are representative of three experiments with similar results. * indicates a *p* < 0.05 when compared to cells that received no stimulus ** indicates a *p* < 0.05 compared to LPS-treated cells without NF-kB inhibitor.

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
