# Peer review of "Lipopolysaccharide from the Cyanobacterium Geitlerinema sp. Induces Neutrophil Infiltration and Lung Inflammation"

_toxins, 2022, doi:10.3390/toxins14040267_

Round 1
Reviewer 1 Report
Throughout, please italicize Geiterinema sp. and also E. coli.
The first time you name E. coli, the genus should be spelled out.
Excellent contribution
Reviewer 2 Report
Manuscript: Cyanobacterium Geitlerinema sp. Lipopolysaccharide induces neutrophil infiltration and lung inflammation
Comment about the title: The title must be revised. “Cyanobacterium Geitlerinema sp. Lipopolysaccharide…” seems incorrect.
General comments: The manuscript highlights the role of cyanobacterial LPS on inducing an inflammatory response in lungs, following cyanobacterial cells inhalation. It is a relevant study that rise the discussion on respiratory diseases following the inhalation of cyanobacterial cells and the effects of cell LPS beyond the well-known cyanotoxins. Also, the authors have made previous research and the actual study represents their next-step in this field of research. However, some writing issues address some inconsistency which must be carefully revised.
I recommend the present manuscript to be reconsidered after major revision.
Please, find bellow the specific comments with more information.
Abstract
It must be rewritten according to the modifications performed in the manuscript.
Introduction
General comment:
The paper highlights the cyanobacterial LPS as a threat for lungs functioning, especially in those people who has asthma. Therefore, it is important firstly to introduce the cyanobacteria as a problem concerning that, besides contaminating drinking water, another relevant exposure route is the inhalation of cells following aerosol formation in rivers, lakes, and even coastal zones with occurrence of CyanoHABs and used for recreational and occupational activities. Subsequently, the authors can focus on the role of those LPS on asthma aggravation and all underlying responses (e.g., inflammatory cascade).
Specific comments:
Lines 52 – 53: This sentence needs complement. The blooms do not result in the symptoms, but being exposed to them by different routes. In addition, it is welcome to write some words concerning exposure routes so that makes sense the presence of cyanobacterial cells, e.g., in the air.
Lines 59 – 61: There is no respiratory distress above mentioned. At least not in the previous paragraph. Please, be more specific.
Furthermore, it is crucial to mention that much has been studied on the toxicological effects of the classical cyanotoxins, especially microcystins. For instance, a previous study has pointed that MC-LR can stimulate alveolar macrophages to produce inflammatory mediators. Hence, it is important to be described before focusing on the LSP which do not depend on the ability of producing MCs or other well-known cyanotoxins.
Line 65: It is important to mention that Geitlerinema is not widely described as a bloom-forming species, despite of a few reports of toxin production. However, here it was used as a model for obtaining cyanobacterial LPS.
Lines 77 – 84: This part is more suitable to be described as a conclusion section or in the last paragraph of the discussion, after resumed.
Results
General comment:
In overall the results are poorly described and with some information which are more suitable to be described in the discussion and methods sections. The authors need to focus on the main goal that is the effect of cyanos LPS on inducing an inflammatory response in mice lungs. All results of experiments and analysis of cellular and biochemical markers must be described with focus on the main goal with the necessary details.
Specific comments:
Lines 87 – 89: This is more suitable to be described in the Discussion.
Lines 89 – 93: This should be described in the Methods.
Line 94: Please, rewrite. This sentence has no sense. E. coli has no lungs.
Lines 97 – 99: This information should be described in the discussion. In addition, why did you only compare Cyano LPS effect with the negative control PBS? It is also a result, e.g., to demonstrate if Cyanos LPS promote a higher inflammatory response in lungs, than that obtained by E. coli LPS.
Lines 101 – 104: Actually, you mean that inflammation cells were evidenced and them determined by hematoxylin-eosin staining, right? Please, clarify!
Also, the way it is written, is more suitable to the Methods section.
Lines 105 – 106: Unfortunately, it is not possible to see any kind of cell in the figures. It is with a very low quality and must be replaced by a figure with a better quality.
Line 108: What does mean "a similar shift"? Did the neutrophils increase or decrease in percentage? Please, be clearer.
Lines 115 – 118: Please, be clearer in the comparisons. If possible, add the statistical data (e.g., p-value).
Line 126 (Figure 1): The graphs and histological figures are with a very low quality. Please, replace. Also, in the graphs add different letters for statistical differences, instead of asterisks.
Line 137: Please, be more specific. Which data in figure 1 can suggest an increase in IL-8?
Lines 145 – 146: This is more suitable to be described in the discussion.
Lines 153 – 154: This finding can be more explored in the discussion. Please, here just describe the obtained results.
Lines 164 – 170: This kind your information is more suitable to be described in the Methods section.
Line 171: Please, replace "expression" by "production" or something that does not refer to gene expression. The way it is written it can lead to a misinterpretation.
Line 173: Again. Please, be careful when using terms that suggest any transcript level analysis.
Lines 181 – 183: More suitable to described in the discussion.
Line 197: Replace "studies" by "model".
Lines 199 – 201: Highlight that cyanos LPS induced an IL-8 production higher than that in induced by E. coli LPS. Also, replace "showed" by "induced".
Lines 201 – 202: Replace this sentence to the discussion section.
Line 213 (Figure 4): Which LPS do you refer in the figures C and D? Cyanos or E. coli one?
Discussion
General comment:
It must be totally rewritten. The way the discussion was constructed looks like an introduction. Please, discuss your results based on the literature. For instance, you have added some sentences along the results that are more suitable to described here in the discussion.
Specific comment:
Lines 226 – 228: Here is a good place to write some words regarding the cyanobacterial genus you have tested as a LPS source. Does Geitlerinema produce any cyanotoxin?
Lines 230 – 233: Also, it is important to mention the combined impact of LPS and cyanotoxins since most of the bloom-forming species are potentially cyanotoxin-producers.
Lines 234 – 241: There are previous studies performed under different exposure routes which assessed the impact of cyanobacterial crude extract and purified cyanotoxins (usually MCs) in mice's lungs. These studies should be mentioned here and compared with your findings.
Lines 243 – 246: These mentioned studies should briefly discussed here.
Lines 247 – 251: Is this a justification for the concentrations of LPS used in the assays? If so, replace this information to the Methods.
Lines 254 – 265: The last paragraph must summarize the main results and bring some conclusion.
Materials and Methods
Line 273: Does this strain has been previously submitted to a cyanotoxins screening? It would be welcome to show any data on cyanotoxins or cyanopeptides production to consider the effect of other substance than LPS.
Line 300: It would be of great relevance to estimate if the LPS dose used here is near to that provoked under exposition to a contaminated aerossol. This highlights an environmentally-relevant model of exposure.
Line 361: One-way ANOVA is only adequate to test data in the figures 1 and 2. For the Figure 4 (data on LPS impact on IL-6 and -8 production), data shown in the Fig.4 A-B should be analyzed by a two-way ANOVA and displayed as bars, instead of lines, since the later indicate a depedence among the replicates. For the data shown in the Fig.4 C-D, the authours should apply an unpared T-test to test any difference between control and LPS treatment.
Please, perform the adequate data analysis acording to the data set and experimental design.
Also, were all the LPS administered in PBS? It is not clear in the methods.
Reviewer 3 Report
Generally, the topic of this manuscript is interesting to me. The writing is clear and easy to follow. It can be acceptable after the following points are justifiably addressed.
- The figures of this manuscript are not clear to read, particularly, these tissue figures cannot be read for comparative analysis.
- Line 92: “E. coli LPS”. The phrase of E.coli should be italicized. This should be done through all the Latin names in the manuscript.
- Line 93-94: “As shown in Figure 1A, the lungs of E. coli LPS showed significant inflammation and cellular infiltrates when compared to mice that were only exposed to PBS”. The author should clearly mention which kind of change occurred in the histology tissue sample indicated the significant inflammation?
- Line 289: “Purity of Geitlerinema sp. LPS was confirmed 289 as previously described”. What is the degree of purity of the LPS prepared in this work? For example, above 80%? This should be mentioned in the manuscript.
- Line 290-291: “To assess the nucleic acid content of the lipopolysaccharide sample, UV measurements were taken on a BioSpec-nano spectrophotometer” Why did the authors specially measure the DNA and RNA content in the purified LPS extracts? Were the nucleic acids not efficiently removed during the purification process?
- Did the authors test the potential effect of the impurity in the prepared LPS on the experimental results? In the line 295, it was mentioned that the nucleic acids accounted for around 10% of LPS extract.
- Line 298-299: “All animal experiments were reviewed and approved by the Institutional Animal Care and Use Committee at Midwestern University” The institutional review board (IRB) approval number should be mentioned here.
- Line 297-305: what is the number of mice tested for each independent experiment? This should be written in the work.
- How many independent replicates were done in this work? This should be mentioned in the work.
Author Response
Please See Atachment

Round 2
Reviewer 2 Report
Overall, the authors have made most of the suggested modifications and improved the manuscript. Also, the arguments and further explanations in response to some suggested modifications were very welcome. Therefore, in my opinion, the manuscript can be accepted after revision of the following minor issues:
- I would like to reinforce that the title must be revised. “Cyanobacterium Geitlerinema sp. Lipopolysaccharide…” seems incorrect.
- In the abstract, the Methods description must be revised.
- Please, perform a general checking in the language and the correct verb tense to describe, e.g., the results and conclusion.
